# Implication of microRNAs in Carcinogenesis with Emphasis on Hematological Malignancies and Clinical Translation

**DOI:** 10.3390/ijms23105838

**Published:** 2022-05-23

**Authors:** Zsuzsanna Gaál

**Affiliations:** Division of Pediatric Hematology-Oncology, Department of Pediatrics, Faculty of Medicine, University of Debrecen, 98 Nagyerdei Krt, 4032 Debrecen, Hungary; gaal.zsuzsanna@med.unideb.hu

**Keywords:** microRNA, acute leukemia, carcinogenesis, differential diagnosis, chemoresistance, prognostic stratification, biomarker, precision oncology, system biology approach

## Abstract

MicroRNAs (miRNAs) are evolutionarily conserved small non-coding RNAs, that are involved in the multistep process of carcinogenesis, contributing to all established hallmarks of cancer. In this review, implications of miRNAs in hematological malignancies and their clinical utilization fields are discussed. As components of the complex regulatory network of gene expression, influenced by the tissue microenvironment and epigenetic modifiers, miRNAs are “micromanagers” of all physiological processes including the regulation of hematopoiesis and metabolic pathways. Dysregulated miRNA expression levels contribute to both the initiation and progression of acute leukemias, the metabolic reprogramming of malignantly transformed hematopoietic precursors, and to the development of chemoresistance. Since they are highly stable and can be easily quantified in body fluids and tissue specimens, miRNAs are promising biomarkers for the early detection of hematological malignancies. Besides novel opportunities for differential diagnosis, miRNAs can contribute to advanced chemoresistance prediction and prognostic stratification of acute leukemias. Synthetic oligonucleotides and delivery vehicles aim the therapeutic modulation of miRNA expression levels. However, major challenges such as efficient delivery to specific locations, differences of miRNA expression patterns between pediatric and adult hematological malignancies, and potential side effects of miRNA-based therapies should be considered.

## 1. Introduction

MicroRNAs (miRNAs) are evolutionarily conserved, single-stranded non-coding RNAs that are 18 to 25 nucleotides in length [1,2]. The first identified miRNA is transcribed from the Caenorhabditis elegans lin-4 locus. The first mammalian miRNA let-7, was discovered seven years later in 2000 [3]. The biogenesis of miRNAs is a multistep process, including the transcription of primary double-stranded transcripts, nuclear processing by the RNase III endonuclease enzyme Drosha, nucleocytoplasmic export by Exportin-5, cytoplasmic cleavage by Dicer, and formation of the RNA-induced silencing complex (RISC) with Argonaute (Ago) proteins [4]. miRNAs are posttranscriptional regulators of gene expression, that function primarily by binding to the 3′ untranslated regions (3′UTR) of their target messenger RNAs (mRNAs), resulting in the inhibition of translation [5,6]. Depending on the microenvironment, miRNAs can target multiple mRNAs, and a single mRNA can also be targeted by several different miRNAs [7]. miRNAs are involved in a wide range of biological processes such as hematopoiesis [8], while dysregulation of miRNA expression levels plays pivotal roles in numerous diseases [6]. The finding that approximately 50 percent of human miRNAs are located at cancer-associated genomic regions (CAGR) suggests that miRNAs are important puzzle-pieces in the pathogenesis of cancer [9].

In this review, causes and consequences of dysregulated miRNA expression patterns are discussed in hematological malignant diseases, including implication of miRNAs in the well-established hallmarks of cancer and widespread interactions of miRNAs with epigenetic modifiers. Special emphasis is placed on the applications of miRNAs in the differential diagnosis and prognostic stratification of acute leukemias. miRNAs are also novel biomarkers for the prediction of chemoresistance, while modulation of miRNA expression levels provides novel therapeutic opportunities and challenges in hematological malignancies.

## 2. Implication of miRNAs in Carcinogenesis and Leukemogenesis

### 2.1. Dysregulation of miRNA Expression Levels in Cancer

The first evidence of miRNA involvement in cancer was published by Calin et al. in 2002, in a study of chronic lymphocytic leukemia (CLL), describing miR-15 and miR-16 as frequently deleted anti-oncomiRs encoded in chromosomal region 13q14 [10]. During the previous twenty years, growing amount of evidence confirmed that expression signatures of miRNAs are associated with types and grades of tumors [11]. Moreover, evaluation of cancer-specific miRNA patterns is supposed to be a more accurate method of classifying cancer subtypes compared to mRNA signatures [12], especially in case of poorly differentiated tumors [13].

Dysregulation of miRNA expression levels is a hallmark of hematological malignancies [12], that is associated with distinct subtypes of acute leukemia, featured by characteristic cytogenetic and molecular genetic alterations. In pediatric acute lymphoblastic leukemia (ALL), seven different subtypes were identified based on miRNA expression profiles, such as down-regulation of miR-127 and miR-143 in precursor B-cell ALL [14]. In patients with acute myeloid leukemia (AML), up-regulation of miR-126 was associated with inv(16)(p13q22) [15], while high expression level of miR-10 was characteristic to the subgroup of NPM1-mutated patients [16]. Altered miRNA expression levels are involved in the pathogenesis of malignant hematological diseases, such as miR-128a and miR-130 were identified in 2021 as downstream targets of MLL-AF4 fusion oncoprotein, that drive the transition from a pre-leukemic stage to acute leukemia in a murine model [17,18]. MiR-192 was confirmed to play a key role in the pathogenesis of CLL through the regulation of Bcl-2, p21, and p53 [19].

According to recent studies, changes in miRNA expression levels are early events during the multistep process of carcinogenesis [20], that are mediated by several different mechanisms (Figure 1a).

Widespread interactions have been confirmed between miRNAs and transcription factors, among which the most well-known is the tumor suppressor p53, controlling the expression of miR-34 family [21,22], while the proto-oncogenic c-Myc transcription factor is implicated in the regulation of miR-17-92 cluster [23]. Disorders of miRNA biogenesis include aberrant promoter methylation of Drosha and Dicer enzymes, that can be a novel biomarker for the early detection of lung cancer [24]. Expression of Drosha was shown to be down-regulated in many types of cancer [25], and the E518K mutation of DGCR8, an RNA-binding protein associated with Drosha, was found in Wilms tumor, resulting in selectively reduced expression of numerous canonical miRNAs [26]. Global inhibition of miRNA biogenesis by depletion of Dicer promoted cell growth in human cancer cell lines [27]. Phosphorylation of Exportin-5 correlated with the global down-regulation of miRNAs and unfavorable prognosis of patients with hepatocellular carcinoma (HCC), while Ago2 protein was found to be up-regulated in various types of cancer [28].

Single-nucleotide polymorphisms (SNPs) are involved in aberrant miRNA biogenesis and disruption of miRNA targets. Drosha rs6877842 and DGCR8 rs417309 SNPs were confirmed to play pivotal roles in carcinogenesis [29]. SNPs disrupting miRNA targets differentiated patients at high risk versus low risk for long-term genitourinary toxicity after prostate cancer radiation therapy [30]. In the let-7 binding site of interleukin 23 receptor (IL23R) mRNA, rs10880677 A > C polymorphism was described as a potential screening biomarker for colorectal cancer (CRC) [31]. rs6426881 at pre-B cell leukemia transcription factor 1 (PBX1) 3′UTR was associated with poor prognosis in breast and gastric cancer by altering the affinity of miR-522-3p to PBX1 [32]. Numerous SNPs within miRNAs have been associated with increased risk of hematological malignancies and with chemotherapy-induced toxicity. SNPs rs12402181 in miR-3117, rs62571442 in miR-3689 and rs10406069 in miR-5196 were associated with increased risk of pediatric ALL [33,34], while rs2114358 in miR-1206 indicated increased risk of chronic myeloid leukemia (CML) [35]. Previous studies suggested that rs2648841 in miR-1208 and rs2114358 in miR-1206 contribute to increased risk of hepatotoxicity and methotrexate-induced oral mucositis in ALL, respectively [36,37].

Epigenetic regulation of miRNAs results in significant alterations of their expression levels and function [38]. Promoter hypermethylation of tumor suppressor miRNAs is a frequent mechanism of gene inactivation in cancer [39]. Silencing of miR-124 by promoter methylation was detected in a wide variety of hematological malignancies including multiple myeloma (MM), ALL, AML, CLL, and non-Hodgkin lymphomas (NHL) [40]. In classic Hodgkin lymphoma, mir-148a was identified as a tumor suppressor miRNA, inactivated by DNA hypermethylation [41], while homozygous methylation of miR-203 was detected in MM cell lines [42].

### 2.2. Implication of miRNAs in the Development of the Hallmarks of Cancer

Altered expression levels of miRNAs are involved in both the initiation and progression of malignant diseases, [13], contributing to all established hallmarks of cancer [43] (Figure 1b). miRNAs play important role in the development of *self-sufficiency in growth signals* and *insensitivity to growth inhibitory signals*. They can act as both oncogenes or tumor suppressors (oncomiRs and anti-oncomiRs, respectively), depending on the tissue where they are expressed [13]. MiR-146a is a dual nature miRNA showing both oncogenic and tumor suppressor properties in different types of leukemias [44]. The first tumor suppressor miRNAs to be established were miR-16-1 and miR-15a, located in the most frequently deleted genomic region (13q14) in CLL [45,46]. Dysregulation of the anti-oncomiR let-7a was confirmed to be essential for sustaining the survival of MLL-rearranged leukemic cells [47]. MiR-20a-5p, miR-203, and miR-939 function as potent tumor suppressors by targeting protein phosphatase 6 catalytic subunit (PPP6C) in AML, modulating JUNB transcription factor in pediatric anaplastic large cell lymphoma (ALCL), and inhibiting cellular proliferation by targeting CREB1 in MM, respectively [42,48,49]. MiR-124 is generally considered as an anti-oncomiR in hematological malignancies [40], while miR-155 acts as a powerful oncomiR in lymphomas and several types of solid tumors by the down-regulation of SHIP1 protein phosphatase [3]. MiR-155 and miR-183 promote cell proliferation by targeting the transcriptional repressor ZNF238 protein in pediatric ALL and regulating programmed cell death 6 (PCD6) protein in pediatric AML, respectively [50,51]. MiR-486-5p is an erythroid oncomiR that cooperates with GATA1 transcription factor in supporting the growth and survival of leukemic cells, contributing to aberrant erythroid phenotype of the megakaryocytic leukemias in Down syndrome [52]. Amplification of the oncogenic miR-21 was registered in breast, hepatocellular, lung, ovarian and prostate cancers [3]. Besides *limitless replicative potential*, cancer cells are also featured by the *evasion of programmed cell death*. In B-cell precursor ALL, up-regulated miR-187-5p induces Bcl-2 protein, resulting in the apoptosis resistance of cells [53], while silencing of miR-127 by promoter hypermethylation contributes to increased expression level of Bcl-6 in bladder cancer [54]. The oncogenic miR-155 is implicated in *sustained angiogenesis* of tumors by down-regulation of von Hippel–Lindau (VHL) tumor suppressor protein [55]. Accumulating evidence indicates that tumor-derived exosomal miRNAs promote angiogenesis, such as miR-210-3p and miR-663b in oral squamous cell carcinoma (OSCC) and cervical cancer, respectively [56,57].

Epithelial-to-mesenchymal transition (EMT) is linked to a variety of cancer-related activities, including *tissue invasion and metastasis* [58]. MiR-338-3p was confirmed to suppress EMT pathway in neuroblastoma cells by targeting matrix metalloproteinase-2 (MMP-2) [59]. Reduced expression of miR-146b-5p in T-cell precursor ALL resulted in the up-regulation of IL17A, which promotes cell migration and invasion [60]. In non-small-cell lung carcinoma (NSCLC) and osteosarcoma cells, the central role for miR-124-3p in carcinoma cell invasion and metastasis has been established [61], while miR-135a enhances metastasis formation in breast cancer by targeting homeobox A10 (HOXA10) protein [62]. Recent evidence suggests the role of miRNAs in *genomic instability* of cancer cells [63]. Low expression level of miR-137 was accompanied by increased frequency of IgH translocations in MM [63], up-regulation of miR-22 and miR-150 increased genomic instability in CML patients by decreasing the activity of an alternative form of nonhomologous end-joining (NHEJ) [64], and increased expression level of miR-186 resulted in chromosomal instability of arsenic-exposed human keratinocytes [65].

miRNAs can be delivered from cancer cells to the *tumor microenvironment* (TME) and vice versa, thereby affecting growth and evolution of transformed cells [66]. Moreover, increased miR-155 and decreased miR-320 expression levels mediate the transformation of normal fibroblasts to cancer-associated fibroblasts (CAFs), providing a stromal framework for cancer cells [3,67]. In step with progression of MM, up-regulation of miR-27b-3p and miR-214-3p promotes apoptosis resistance of bone marrow fibroblasts [68]. Antitumor effect of miR-340 in pancreatic ductal adenocarcinoma is exerted by promoting macrophages to acquire M1-like phenotype [69], while exosomal delivery of oncomiRs such as miR-126, miR-144 and miR-155 from breast cancer cells to TME adipocytes results in their conversion into cancer-associated adipocytes [66]. In T-cell precursor ALL, increased level of miR-29b was confirmed to decrease cytotoxicity by natural killer (NK) cells, thereby contributing to the escape of leukemic cells from immune surveillance [70].

### 2.3. miRNAs and Metabolic Reprogramming in Leukemia

Aberrant nature of cancer metabolism was first recognized by Otto Warburg in the early 1920s when he postulated that tumor tissues have an increased rate of glucose uptake [71]. Besides elevated rate of glycolysis, cancer cells undergo complex metabolic reprogramming, including increased glutamine consumption and enhanced anabolism. TME cells are also featured by active metabolism, resulting in the suppression of antitumor immune response [72]. Cancer metabolic networks control not only energy balance and growth of tumor cells but are also important modulators of epigenetic mechanisms [71]. Leukemic progenitors activate different metabolic programs in association with genetic alterations and the levels of their differentiation blockade [73]. Metabolic alterations of leukemic cells play a role in leukemogenesis and have an impact on prognosis [73]. In serum samples of patients with AML, significant metabolomic changes have been detected [74]. Glutamine level was elevated among patients with French-American-British (FAB) subgroups M4 and M5, the level of alanine was higher in FLT3-ITD positive patients, and significant decrement of glutamate and phosphocoline levels were registered in the NPM1-mutated subgroup of patients [73]. Comparing FLT3-ITD positive and FLT3 wild-type pediatric AML patients, differential abundance of 21 metabolites in plasma and 33 metabolites in leukemic cells have been detected [75]. In the cytogenetically normal subgroup of adult AML patients (CN-AML), a panel of six metabolites (lactate, 2-oxoglutarate, pyruvate, 2-hydoxyglutarate, glycerol-3-phosphate, citrate) was able to independently assess prognostic potential [76].

In recent years, implication of miRNAs in the metabolic reprogramming of tumor cells has been confirmed, including the regulation of glycolysis, glutaminolysis and anabolic pathways, such as miR-210, miR-23a, and miR-33a, respectively [77,78]. According to the latest data, impaired expression of miR-652-5p leads to decreased levels of ATP, lactate, and pyruvate in T-ALL cell lines, thereby providing a potential drug target [79]. A special subgroup of miRNAs, so-called mitomiRs, are localized in the mitochondria and play an essential role in the control of cancer cell metabolism by targeting key transporters, metabolic enzymes and several oncogenic pathways [80]. Expression levels of metabolism-regulating miRNAs in leukemic stem and progenitor cells have to be further evaluated.

### 2.4. Interactions between miRNAs and Epigenetic Regulatory Circuits

miRNAs are involved in the regulation of those epigenetic mechanisms that play an important role in early stages of leukemogenesis [81,82]. According to a prominent body of evidence, these so-called epi-miRNAs are implicated in the regulation of both DNA-methylation and histone modification [82,83]. 

In AML, miR-29b induces global DNA hypomethylation and re-expression of tumor suppressor genes by targeting DNA methyltransferase (DNMT) enzymes [84]. In estrogen receptor (ER)-positive breast cancer, miR-29c-5p was found to trigger early abnormal DNA-methylation through direct targeting of DNMT3A [85]. DNMT1 enzyme was identified as a target of miR-152 [86], and miR-137 inhibits the invasion and metastasis of nasopharyngeal cancer cells by regulating the histone lysine demethylase enzyme LSD1 [58]. In breast cancer, miR-22 was identified as an oncomiR that targets the methylcytosine dioxygenase TET2 enzyme and promotes cell migration [87]. The potent tumor suppressor role of miR-377 in osteosarcoma is associated with the inactivation of histone acetyltransferase 1 (HAT1)-mediated Wnt signaling pathway [88]. In Waldenström macroglobulinaemia, the HAT enzyme KAT6A is down-regulated by miR-206-3p, while histone deacetylase (HDAC) enzymes HDAC4 and HDAC5 are targeted by miR-9-3p [89]. Cancer cell proliferation and metastasis in HCC were found to be suppressed by miR-20a through the regulation of the histone methyltransferase enzyme EZH1 [90].

## 3. Regulation of Hematopoiesis by miRNAs

Lineage- and maturation-specific expression of miRNAs in hematopoietic cells was first reported by Chen and colleagues in 2004 [91]. Since then, numerous miRNAs have been confirmed to interact with transcription factors crucial for normal development of hematopoietic progenitor cells [38]. Early stages of hematopoiesis, stem cell maintenance, lineage selection, and terminal differentiation of hematopoietic cells are all affected by miRNAs that function as downstream effectors of transcription factors [38,92].

In hematopoietic stem cells (HSCs), high expression of miR-125b was described [81], while enforced expression of this miRNA resulted in the blockade of granulocytic differentiation in murine 32D cells [93]. MiR-223 is a fine-tuner of granulocytic differentiation [94], regulated by NFIA and C/EBPα transcription factors [95]. MiR-150 and miR-486-3p were identified as critical factors in the lineage selection of megakaryocyte-erythrocyte progenitors, by targeting MYB and c-MAF transcription factors, respectively [92,94]. While miR-150 was preferentially expressed in megakaryocytic lineage [94], enforced expression of miR-486-3p increased erythropoiesis and blocked megakaryocytopoiesis in hematopoietic stem and progenitor cells [92]. MiR-17-5p, miR-20a, miR-106a, and miR-129 inhibit monocytopoiesis by repressing RUNX1 transcription factor [92]. FOXP1, a transcription factor that is required for early B-cell development, was identified as a key target of miR-34a [96]. Besides miR-34 family, the miR-17-92 cluster, miR-21-5p, miR-29 family, miR-125b-5p, miR-150-5p, miR-155-5p, and miR-181 also appeared as important regulators of B-cell development [38]. MiR-17-92 promotes early differentiation of T cells [97], and miR-150 is implicated in T lymphopoiesis through the regulation of NOTCH3 transcription factor [98]. Development of NK cells is regulated by a wide range of miRNAs including miR-155 and miR-181a [99].

## 4. Clinical Applications of miRNAs Focusing on Hematological Malignancies

### 4.1. Differential Diagnosis and Early Detection of Hematological Malignancies

Alterations of miRNA expression levels can be applied as non-invasive biomarkers to make advances in both differential diagnosis and early detection of hematological malignancies (Table 1). Based on the expression signature of four miRNAs (miR-128a, miR-128b, let-7b, miR-223), ALL and AML can be distinguished with an accuracy rate of >95 percent [100]. MiR-200c and miR-326 could serve as diagnostic biomarkers for pediatric ALL [101], while a panel of 16 miRNAs may differentiate between pediatric T-ALL and B-ALL, among which miR-29c-5p was identified as the best discriminator [102]. According to recently published data, the expression ratio of miR-92a/miR-638 in blood plasma could be applied as a novel biomarker for AML [103]. In pediatric AML, levels of miR-25, miR-155, miR-196b, and miR-370 are potential diagnostic biomarkers [104,105], among which miR-196b is specific for FAB subgroups M4 and M5 [104].

While miR-32-5p, miR-98-5p, miR-145-5p, miR-185-5p, miR-192, and miR-374b-5p are potential biomarkers for the early detection of CLL [19,106,107], a miRNA expression signature composed of 13 genes can differentiate between subgroups of CLL patients, based on the level of ZAP-70 expression and immunoglobulin heavy chain variable region gene (IgHV) mutational status [5]. MiR-451 was found to be up-regulated in plasma samples of CML patients in the chronic phase [108], while miR-126, miR-155, and miR-222 were up-regulated in samples obtained during blast crisis [109].

The expression of let-7f, miR-9, and miR-27a can serve as biomarkers to distinguish classic Hodgkin lymphoma from other B-cell lymphomas [110]. Activated B-cell and germinal center-like diffuse large B-cell lymphoma (DLBCL) cell lines can be differentiated based on a panel of nine miRNAs [111]. In lymph gland samples, a set of 20 miRNAs were identified as judgmental indicators to differentiate between reactive lymphoid hyperplasia (RLH) and peripheral T-cell lymphoma, not otherwise specified (PTCL-NOS) [112]. The expression levels of miR-142-3p, miR-155, and miR-203 might be helpful biomarkers for the differential diagnosis between chronic gastritis and gastric MALT lymphomas [113]. The level of miR-30c in the cerebrospinal fluid (CSF) can be applied to differentiate between patients with primary lymphomas of the central nervous system (CNS) and secondary spread of systemic lymphoma to the CNS [114].

In MM, miR-4254 was found to be the most promising diagnostic biomarker [115]. In addition, a combination of up-regulated miR-34a and down-regulated let-7e could distinguish MM from both control and monoclonal gammopathy of undetermined significance (MGUS) with high sensitivity and specificity [116].

### 4.2. Advances in Prognostic Stratification of Hematological Malignancies

Growing number of miRNAs has been identified as novel biomarkers for prognostic stratification in hematological malignancies, contributing to optimization of therapy in numerous subgroups of patients (Table 1).

While the overall survival (OS) rate of ALL patients was better with low expression of miR-429 [117] and high expression level of miR-16 was also found to be a good prognostic factor [118], up-regulation of miR-146a, promoter hypermethylation of miR-124 and low expression levels of miR-99a and miR-100 correlated with poor prognosis in adult ALL patients [44,118,119].

In pediatric ALL, miR-10a, miR-134, and miR-214 were linked with a favorable outcome arising from their tumor suppressor activity [14]. Increased expression of miR-155 and down-regulation of miR-99a and miR-708 were associated with poor prognosis in childhood ALL [50,120], and the high degree of miR-152 promoter methylation indicated poor clinical outcome among t(4;11)-positive infant ALL patients [86]. Up-regulation of miR-155 and miR-181a was described in pediatric ALL patients with high levels of minimal residual disease (MRD) [121]. In pediatric precursor B-cell ALL, down-regulation of miR-151-5p and miR-451, up-regulation of miR-1290, or a combination of all three predicted inferior relapse-free survival (RFS) [122]. Increased expression of miR-143 and/or miR-182 at the end of induction treatment was associated with significantly higher risk for short-term relapse and death [123], while a set of six miRNAs (miR-101-3p, miR-4774-5p, miR-1324, miR-631, miR-4699-5p, and miR-922) was confirmed to be up-regulated in early relapse in pediatric B-ALL [124].

In AML patients, higher expression of the miR-181 family was reported to positively correlate with favorable prognosis [19,125]. Up-regulation of miR-195 was associated with favorable outcome in the CN-AML group [126]. Down-regulation of miR-504-3p and high expression levels of miR-363, miR-191 and miR-199a were related to poor prognosis, shorter event-free survival (EFS), and worse OS data in AML patients [81,127,128].

Low expression of miR-193b-3p and increased levels of miR-509 and miR-542 were found to be poor prognostic factors, whereas up-regulation of miR-146a and miR-3667 were linked with favorable outcome in pediatric and adolescent AML [129,130]. Pediatric AML patients with reduced levels of miR-29a and miR-370 had shorter RFS and OS [19,131], while down-regulation of miR-199a correlated with shorter EFS [132].

In CLL, down-regulation of miR-29b, miR-29c, and miR-181b were associated with unfavorable prognosis [8,133]. In Egyptian CLL patients, increased expression levels of miR-18a, miR-19b-1, and miR-92a-1 also had an adverse prognostic value [134]. Up-regulation of miR-650 and miR-708 was associated with favorable prognosis in CLL [135].

In CML, decreased expression of miR-320a was associated with worse OS and RFS rates [136], patients with early treatment response (ETR) had significantly higher levels of miR-150 [137], while up-regulation of miR-486-5p was associated with favorable prognosis [138].

In DLBCL, low expression levels of miR-27b and miR-129-5p were linked with poor clinical outcome [139,140]. Up-regulation of miR-130a was associated with adverse prognosis of primary gastrointestinal DLBCL [141], high expression of miR-22 correlated with a worse progression-free survival (PFS) [142], whereas increased expression levels of miR-199a and miR-497 indicated favorable outcome in DLBCL patients [143]. MiR-34a-5p was identified as a novel biomarker for the transformation of gastric MALT lymphoma to gastric DLBCL [144]. Down-regulation of miR-7 and miR-223 was associated with inferior OS data in NHL and mantle cell lymphoma, respectively [145,146].

Low levels of let-7e, miR-223-3p and miR-744 were associated with shorter remission and OS in MM patients [147,148], while up-regulation of miR-720 and miR-1246 correlated with shorter PFS data [149].

### 4.3. Implication of miRNAs in Chemosensitivity, Prediction of Chemoresistance in Hematological Malignancies

Drug resistance is one of the topmost challenges of cancer chemotherapy, caused by many mechanisms including decreased intake or increased release of drugs, degradation or deactivation, target modification, enhanced DNA damage repair activity, and alterations of cell cycle checkpoints [21,150]. Altered miRNA expression levels have been associated with chemoresistance in hematological malignancies (Table 1).

Up-regulation of miR-142-3p and the miR-17-92 cluster in B-ALL was associated with acquired resistance to dexamethasone [151], while transfection of Jurkat cells with miR-34a resulted in increased sensitivity to doxorubicin [152]. MiR-206 is implicated in chemoresistance of B-lymphoblasts by targeting the guanine nucleotide exchange factor NET1 [153]. In pediatric ALL, resistance to daunorubicin and vincristine was characterized by the up-regulation of miR-99a, miR-100, and miR-125b [14]. According to recently published results, good and poor prednisone response can be distinguished by the evaluation of a miRNA signature composed of 8 genes (miR-18a, miR-193a, miR-218, miR-532, miR-550, miR-625, miR-633, and miR-638) [154]. Increased expression of miR-221 and miR-21 correlated with poor response to induction therapy in T-ALL and B-ALL, respectively [155,156]. Down-regulation of miR-326 was introduced as a biomarker for drug resistance in childhood ALL, showing inverse correlation with the expression level of ABCA2 transporter [101]. A negative correlation was also identified between the expression of two miRNAs (miR-324-3p and miR-508-5p) and ABCA3 transporter, that is associated directly with chemoresistance [157].

Decreased expression of miR-874-3p and up-regulation of miR-15a-5p and miR-21-5p contribute to the development of chemoresistance to cytosine arabinoside in AML and to doxorubicin in CN-AML, respectively [158,159]. MiR-143 sensitizes AML cells to cytosine arabinoside via targeting autophagy-related proteins ATG2B and ATG7 [160]. MiR-15a-5p induces resistance, whereas miR-9 enhances sensitivity of AML cells to daunorubicin through the abrogation of autophagy and by targeting the EIF5A2/MCL-1 axis, respectively [161,162]. MiR-217 enhances chemosensitivity of AML cells to doxorubicin by targeting KRAS [163], while miR-204 potentiates the sensitivity of acute promyelocytic leukemia (APL) cells to arsenic trioxide (ATO) [164]. MiR-29a and miR-100 were identified as significant predictors of chemotherapy response in pediatric AML [165]. Up-regulation of miR-125b indicated increased drug resistance in pediatric APL [166].

In CLL, fludarabine resistance was associated with increased expression levels of miR-21, miR-148a, miR-155, and miR-222 [8,167]. On the other hand, increased expression of miR-181a was found to sensitize CLL cells to fludarabine [168]. MiR-9 was confirmed to play a critical role in the development of multidrug resistance in CML by targeting ABCB1 transporter [169]. Down-regulation of miR-142-5p, miR-199b, miR-217, miR-221, and miR-365a-3p correlated with resistance to tyrosine kinase inhibitors (TKIs) in CML [8,170,171]. Measurement of circulating miR-146a and miR-451 expression can also predict treatment response to TKIs [172,173]. MiR-577 promotes sensitivity of CML cells to imatinib by targeting the NUP160 subcomplex of the nuclear pore [174].

In DLBCL, high expression of miR-155 and down-regulation of miR-193b-5p and miR-1244 were associated with rituximab plus cyclophosphamide, doxorubicin, vincristine, and prednisone (R-CHOP) treatment failure [175,176]. Overexpression of miR-125b-5p sensitized cutaneous T-cell lymphoma cells to bortezomib via modulation of MAD4 protein [21], while miR-223-3p is implicated in ibrutinib resistance through regulation of the conserved helix-loop-helix ubiquitous kinase (CHUK)-NFκB signaling pathway in mantle cell lymphoma [177].

Down-regulation of miR-155 and miR-145-3p was associated with bortezomib resistance [178,179], while up-regulation of miR-221 and 222 was found to inhibit autophagy, thereby promoting dexamethasone resistance in MM cells [180].

### 4.4. Changes of miRNA Expression Levels during Chemotherapy

The first report to show that chemotherapy influences miRNA expression in human cancer cells was published in 2006, discussing the changes of miRNA profiles in cholangiocarcinoma cells during gemcitabine treatment [181]. Growing number of evidence suggests that chemotherapy causes significant alterations of miRNA expression profiles in solid tumors and hematological malignancies. In ALL cell lines, increased levels of miR-15a and miR-16-1 were registered after prednisolone treatment [182]. In ALL patients, decreased expression level of miR-146a and up-regulation of let-7e was detected after 14 days of treatment [44,183]. Obinutuzumab-induced CD16 stimulation and bortezomib led to an up-regulation of miR-155-5p and miR-29b in follicular lymphoma and in murine models of cutaneous T-cell lymphoma, respectively [184,185]. In CML, dasatinib affected the expression of miR-let-7d, miR-let-7e, miR-15a, miR-16, miR-21, miR-130a and miR-142-3p, while imitanib was shown to modulate miR-15a, miR-21, miR-122, miR-126, and miR-130a levels [186,187,188].

### 4.5. miRNA-Based Anticancer Therapeutic Approaches

The Nobel Prize in Medicine was awarded to Andrew Z. Fire and Craig C. Mello in 2006 for the discovery of RNA interference, gene silencing by double-stranded RNA molecules [189]. The first miRNA-targeted drug was miravirsen (SPC3649), a locked nucleic acid (LNA)-modified antisense oligonucleotide targeting miR-122, that is currently in phase II clinical trial for the treatment of hepatitis C virus infection [190,191]. MRX34 (miRNA Therapeutics), containing a double-stranded miR-34a mimic encapsulated in ionizable liposomes, was the first miRNA mimic to enter a phase I clinical trial in 2013 (NCT01829971) in several solid tumors and hematological malignancies [3,192]; however, the trial was closed due to severe immune-related adverse events [6].

Modulation of miRNA expression levels is possible with both mimic and inhibitor molecules. Increased stability, longer half-lives, and higher efficiency are aimed by the optimization of delivery and chemical modifications of oligonucleotide molecules [3].

In AML cell lines THP-1 and U937, overexpression of miR-20a-5p suppressed cell proliferation, induced cell cycle arrest and apoptosis, representing a novel therapeutic target for AML [48]. Overexpression of miR-150 in NK/T-cell lymphoma cells resulted in substantially enhanced sensitivity to ionizing radiation treatment [193], while miR-28-5p was found to suppress the growth of DLBCL cells by inhibiting the expression of YWHAZ protein [194]. In animal models, targeted delivery of miR-200c with nanoparticles suppressed proliferation of triple-negative breast cancer cells [195]. Up to recently published results, miR-15 and miR-16 mimics could be used for the therapy of Bcl-2-overexpressing tumors [45]. Delivery of miR-16 using an EnGeneIC Delivery Vehicle (EDV) nanocell system targeting epidermal growth factor receptor (EGFR) in NSCLC and malignant pleural mesothelioma resulted in significant tumor reduction [196]. In NSCLC and breast cancer cells, overexpression of miR-20b-5p combined with pembrolizumab potentiated sensitivity of tumor cells to radiotherapy by repressing PD-L1/PD1 [197].

miRNA sponges are synthetic RNA molecules containing multiple high affinity miRNA binding sites to reduce the abundance of oncogenic miRNAs within the cell [62]. Overexpression of the long non-coding RNA molecule, MIR17HG promoted homoharringtonine (HHT)-induced apoptosis of AML cells by sponging miR-21, resulting in the up-regulation of the tumor suppressor PTEN protein phosphatase [198]. MiR-10b sponge was confirmed to inhibit metastasis formation in breast cancer by the up-regulation of HOXD-10 [199]. Silencing of miR-146a, miR-155, miR-181a, and let-7e enhanced the effect of prednisolone treatment in pediatric ALL [200], while blockade of let-7a-5p may be a novel therapeutic strategy in APL through the activation of caspase-3 [201]. Inhibition of miR-182 by the transfection of LNA-anti-miR-182 resulted in decreased proliferation of APL cells through the modulation of CASP9 expression [202].

Besides the modulation of their expression levels by mimics and inhibitors, miRNAs can be applied in anticancer treatment by further approaches including miRNA-level based therapeutic decisions and epigenetic drugs restoring altered miRNA expression levels. In AML, patients with high expression levels of miR-363 may be highly recommended for early allo-HSCT regimen [128]. Demethylation treatment contributes to the restoration of tumor suppressor miRNAs. Tumor-specific promoter hypermethylation and silencing of miR-124 can be reversed by the hypomethylating agent 5-aza-2-deoxycytidine, that is associated with consequent down-regulation of CDK6 enzyme [40]. In the t(4;11)-positive B-ALL cell line SEMK2, expression levels of seven miRNAs (miR-10a, miR-152, miR-200a, miR200b, miR429, miR-432, and miR-503) increased following demethylation by the cytidine analog Zebularine [86]. In MM cell lines, 5-aza-2-deoxycytidine led to promoter demethylation and re-expression of miR-203 [42]. HDAC inhibitors (HDACi) have also been confirmed to modify miRNA expression levels. Sapacitabine induced the expression of miR-182 in primary AML blasts, thereby sensitizing AML cells to DNA-damaging agents [203]. In synergism with imatinib mesylate, vorinostat was found to elicit marked inhibition of CML stem cells by the up-regulation of miR-196a [204]. Pan-HDACIs vorinostat and panobinostat inhibit metastasis formation in advanced cutaneous T-cell lymphoma by restoring the tumor suppressor miR-150 [205]. In gastric cancer, panobinostat increased the expression of the tumor suppressor miR-874, thereby inhibiting proliferation of tumor cells [206].

### 4.6. Clinical Applications of Exosomal miRNAs

Exosomes are highly stable endosomal sorting pathway products with a diameter of 30–100 nm. Accumulating evidence highlights their roles in the pathogenesis of cancer, and numerous exosomal miRNAs in serum have been identified as novel biomarkers for early detection and prognosis prediction of malignant diseases [207]. Furthermore, exosomal miRNAs derived from cancer cells modulate chemosensitivity and TME, providing promising therapeutic targets [11].

In 2015, Hornick et al. developed a panel of three exosomal miRNAs (miR-150, miR-155, and miR-1246) as a minimally invasive early biomarker of MRD in AML [208]. Exosomal miR-7-5p and miR-425-5p, derived from bone marrow mesenchymal stem cells (BM-MSCs), were confirmed to inhibit proliferation, and promote apoptosis of AML cells by targeting PI3K/AKT/mTOR signaling pathway and Wilms tumor 1-associated protein (WTAP), respectively [209,210]. Exosomal miR-125-5p reduced sensitivity of DLBCL cells to rituximab treatment [211], while increased levels of exosomal miR-99a-5p and miR-125b-5p in sera samples of DLBCL patients were associated with shorter PFS data [212]. In MM patients, down-regulation of four exosomal miRNAs (miR-5a-5p, miR-16-5p, miR-17-5p and miR-20a-5p) was registered in the bortezomib resistance group [207], and high levels of exosomal miR-1305 indicated poor OS [213]. MiR-221 and miR-222 secreted by CRC cells are implicated in the formation of a hospitable metastatic environment in the liver [214].

## 5. Concluding Remarks and Future Perspectives

Resulting from advances in molecular diagnostic evaluation, chemotherapeutic protocols and supportive treatment of patients, survival outcomes in hematological malignancies have considerably improved during the previous decades. The 5-year OS rate for pediatric acute leukemia is now more than 90% in ALL and 60–70% in AML [12]. Chemoresistance and toxicity are still among the major causes of treatment failure and mortality [215]. According to recent evidence, identification of new biomarkers can be utilized in early identification, differential diagnosis, non-invasive MRD monitoring, and prognostic stratification of acute leukemias [81].

miRNAs are a class of highly stable, non-coding small RNAs that can be easily quantified in body fluids and paraffin-embedded tissues [38]. Besides the high number of currently ongoing studies to assess their potential as MRD biomarkers [206], miRNAs are novel candidates for therapeutic intervention [216]. However, modulation of miRNA levels holds great opportunities, major challenges must be overcome to provide specificity and avoid side effects including activation of innate immune system and undesirable toxicities [217]. Resulting from the great numbers of their targets and the complexity of the regulatory network that determines their levels, the biological impact of a certain miRNA expression pattern is highly dependent on the tissue microenvironment. Since the biological behavior of pediatric malignancies is markedly different from their adult counterparts [218], it is not surprising that major differences between miRNA expression signatures of pediatric and adult acute leukemias have been described [19].

Based on the complex pathogenesis of acute leukemias and recent innovation in molecular genetic diagnostics and bioinformatics, major focuses of onco-hematology include precision oncology and system biology approach, aiming the clinical translation of novel biomarkers and personalized treatment opportunities. Application of miRNAs in the early detection, prognostic stratification and therapy of hematological malignancies can contribute to more efficient and less toxic antileukemic treatment, thereby improving survival and life quality of patients.

## Figures and Tables

**Figure 1 ijms-23-05838-f001:**
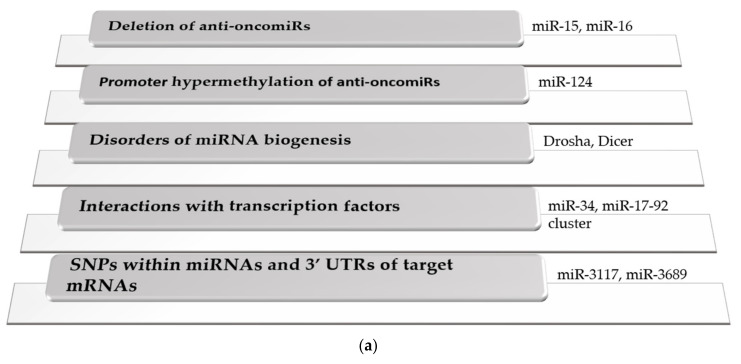
(**a**) Mechanisms of dysregulated miRNA expression in hematological malignancies. Abbreviations: SNP: single nucleotide polymorphism, UTR: untranslated region. (**b**) Implication of miRNAs in the development of the hallmarks of cancer.

**Table 1 ijms-23-05838-t001:** Clinical applications of miRNAs in hematological malignancies. See references in the text.

	Differential Diagnosis	Prognostic Stratification	Chemoresistance
**ALL**	miR-128a, miR-128b, let-7b, miR-223	miR-146a, miR-429, miR-124 hypermethylation, miR-99a, miR-100, miR-16	miR-206, miR-34a, miR-142-3p, miR-17-92 cluster
**Pediatric** **ALL**	miR-29c-5p, miR-326, miR-200c	miR-10a, miR-134, miR-214, miR-708, miR-99a, miR-151-5p, miR-451, miR-1290, miR-155, miR-181a, miR-143, miR-182, miR-152 promoter methylation, miR-101-3p, miR-4774-5p, miR-1324, miR-631, miR-4699-5p, miR-922	miR-125b, miR-99a, miR-100, miR-324-3p, miR-508-5p, miR-18a, miR-532, miR-218, miR-625, miR-193a, miR-638, miR-550, miR-633, miR-21, miR-326, miR-221
**AML**	miR-128a, miR-128b, let-7b, miR-223, miR-92a/miR-638 ratio	miR-181 family, miR-504-3p, miR-191, miR-199a, miR-195, miR-363	miR-874-3p, miR-15a-5p, miR-21-5p, miR-9, miR-217, miR-143, miR-204
**Pediatric** **AML**	miR-196b, miR-155, miR-25, miR-370	miR-193b-3p, miR-370, miR-29a, miR-509, miR-542, miR-146a, miR-3667, miR-199a	miR-29a, miR-100, miR-125b
**CLL**	miR-192, miR-32-5p, miR-98-5p, miR-374b-5p, miR-145-5p, miR-185-5p	miR-181b, miR-650, miR-708, miR-29b, miR-29c, miR-18a, miR-19b-1, miR-92a-1, miR-17	miR-148a, miR-222, miR-21, miR-181a, miR-155
**CML**	miR-451, miR-222, miR-126, miR-155	miR-486-5p, miR-320a, miR-150	miR-217, miR-199b, miR-221, miR-577, miR-451, miR-146a, miR-9, miR-142-5p, miR-365a-3p
**Lymphomas**	let-7f, miR-9, miR-27a, miR-142-3p, miR-155, miR-203, miR-30c	miR-130a, miR-199a, miR-497, miR-34a-5p, miR-22, miR-129-5p, miR-27b, miR-7, miR-223	miR-125b-5p, miR-155, miR-1244, miR-193b-5p, miR-223-3p
**MM**	miR-34a, let-7e, miR-4254	miR-223-3p, miR-744, let-7e, miR-720, miR-1246	miR-145-3p, miR-155, miR-221, miR-222

## Data Availability

No new data were created or analyzed in this study. Data sharing is not applicable to this article.

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
