# Peer review of "Implication of microRNAs in Carcinogenesis with Emphasis on Hematological Malignancies and Clinical Translation"

_ijms, 2022, doi:10.3390/ijms23105838_

Round 1

Reviewer 1 Report

Gaal summarised the roles of miRs in haematological cancers. It is a good summary however there is nothing new. There is no new conclusions, no new clinical/mechanistic hypothesis. Dr Gaal is the sole author which is not common for extensive reviews also Dr Gaal published only 2 miR related research papers, one is an animal model. This review is not suitable for Cancers as there is nothing novel.

miRs should be written as miRs not MiRs

The figures or the basic drawings should be improved especially contrast should be increased.

Table is highly basic, no detailed discussion on specific malignancy/patient info/metastasis.

Author Response

Thank you very much for reviewing the manuscript. Please find below my answers to the comments.

I absolutely agree with the Reviewer that my previous research expertise in the field is very limited. It is a great challenge for me and my colleagues at the Department of Pediatric Hematology-Oncology to perform successful research programs in the future, contributing to novel experimental results and conclusions. 

The kind invitation to this special issue provided me a unique opportunity to summarize the latest results and widen my knowledge in the field. Although the manuscript focuses on miRNAs, it also highlights the complexity of epigenetic regulatory circuits and metabolic reprogramming of cancer cells. MiRNAs are implicated in these novel hallmarks of hematological malignancies, opening up a new horizon for clinical translation.

miRs are written instead of MiRs in the revised version of the manuscript.

Figures are included in the revised version with increased contrast, and formatting of Table1 has been modified. Groups of miRNAs, implicated in prognosis and chemoresistance, have been distinguished based on the age of patients (pediatric or adult) in case of ALL and AML. Specific types of hematological malignancies are featured by different miRNA expression signatures, that can be utilized in both differential diagnosis and prognostic stratification (included in Table 1).

Reviewer 2 Report

I really enjoyed going through the reviews entitled “Implication of microRNAs in carcinogenesis with emphasis on 2 hematological malignancies and clinical translation” by Dr. Zsuzsanna. I believe this will be a great compilation of the existing literature and will certainly be enjoyed by the scientific community.

Author Response

I would like to express my gratefulness for reviewing the manuscript and the positive feedback. Spell check has been performed and I hope that the manuscript will meet the high standards of IJMS.

Reviewer 3 Report

This is a fairly well-written review of the microRNAs in hematological malignancies. However, there are a number of excellent reviews already published on microRNAs, albeit not on hematological cancers. 

The information presented is sound.

Author Response

Here I would like to express my gratefulness for reviewing the manuscript and the positive feedback. I hope that preparation of this manuscript can be utilized also at our institute for planning and performing novel experimental research projects in the field.

Reviewer 4 Report

In the article, "Implication of microRNAs in carcinogenesis with emphasis on hematological malignancies and clinical translation",  the authors presents a thorough overview of the evidence supporting the role of microRNAs as a biomarker for hematological disorders. There are a few small issues with this review that require attention before it can be published. Comments can be found below:

1) Its a small thing, but I dont think the word "review" should be capitalized. Please double check this. 

2) Abstract - the sentence starting "Growing number..." This sentence is a little awkward in how its constructed. Perhaps a word has been accidentally omitted.

 3) Introduction - the sentence starting - "MiRNAs are considered..." This sentence is a little awkward as well. 

4) MiRNAs and metabolic reprogramming in leukemia - the sentence starting - "Up to the latest published..." Use of the phrase "up to the latest.." is awkward and perhaps needs to be replaced with "according to".

Author Response

Thank you very much for reviewing the manuscript.

Please find below my answers for the comments:

  1. The word "review" is not capitalized in the revised version of the manuscript.
  2. The sentence has been modified: "Synthetic oligonucleotides and delivery vehicles aim the therapeutic modulation of miRNA expression levels."
  3. The sentence has been modified: "miRNAs are involved in a wide range of biological processes such as hematopoiesis, while the dysregulation of miRNA expression levels plays pivotal roles in numerous diseases."
  4. The phrase "up to" has been replaced with "according to" (lines 205-207 of the revised manuscript)  

Round 2

Reviewer 1 Report

I cannot see what was changed as it was not highlighted. I still think the same, there is nothing new this review brings to the field. No mechanism, no hypothesis.